# *Plectranthus zeylanicus*: A Rich Source of Secondary Metabolites with Antimicrobial, Disinfectant and Anti-Inflammatory Activities

**DOI:** 10.3390/ph15040436

**Published:** 2022-03-31

**Authors:** Mayuri Napagoda, Jana Gerstmeier, Hannah Butschek, Sybille Lorenz, Sudhara De Soyza, Mallique Qader, Ajith Nagahawatte, Gaya Bandara Wijayaratne, Bernd Schneider, Aleš Svatoš, Lalith Jayasinghe, Andreas Koeberle, Oliver Werz

**Affiliations:** 1Department of Biochemistry, Faculty of Medicine, University of Ruhuna, Galle 80000, Sri Lanka; sudhara.gamini@yahoo.com; 2Department of Pharmaceutical/Medicinal Chemistry, Institute of Pharmacy, Friedrich Schiller University Jena, Philosophenweg 14, D-07743 Jena, Germany; jana.giesel-gerstmeier@helios-gesundheit.de (J.G.); hannah_butschek@gmx.de (H.B.); andreas.koeberle@uibk.ac.at (A.K.); 3Research Group Mass Spectrometry and Proteomics, Max Planck Institute for Chemical Ecology, Hans-Knoell-Strasse 8, D-07745 Jena, Germany; lorenz@ice.mpg.de (S.L.); svatos@ice.mpg.de (A.S.); 4Natural Products Research Division, National Institute of Fundamental Studies, Hantana Road, Kandy 20000, Sri Lanka; mallique.qader@gmail.com (M.Q.); ulbj2003@yahoo.com (L.J.); 5Department of Microbiology, Faculty of Medicine, University of Ruhuna, Galle 80000, Sri Lanka; ajithnagahawatte@yahoo.co.uk (A.N.); gayabw@yahoo.co.uk (G.B.W.); 6Research Group Biosynthesis/NMR, Max Planck Institute for Chemical Ecology, Hans-Knoell-Strasse 8, D-07745 Jena, Germany; schneider@ice.mpg.de; 7Michael Popp Institute and Center for Molecular Biosciences Innsbruck (CMBI), University of Innsbruck, Mitterweg 24, A-6020 Innsbruck, Austria

**Keywords:** anti-inflammatory, antimicrobial, disinfectant, *Plectranthus zeylanicus*

## Abstract

*Plectranthus zeylanicus* Benth is used in Sri Lankan folk medicine as a remedy for inflammatory conditions and microbial infections. Our previous investigations revealed potent 5-lipoxygenase (5-LO) inhibitory activity in lipophilic extracts of this plant, supporting its anti-inflammatory potential. In-depth studies on the antimicrobial activity have not been conducted and the bioactive ingredients remained elusive. As a continuation of our previous work, the present investigation was undertaken to evaluate the antimicrobial activity of different extracts of *P. zeylanicus* and to isolate and characterize bioactive secondary metabolites. Different organic extracts of this plant were analyzed for their antibacterial activity, and the most active extract, i.e., dichloromethane extract, was subjected to bioactivity-guided fractionation, which led to the isolation of 7α-acetoxy-6β-hydroxyroyleanone. This compound displayed strong antibacterial activity against methicillin-resistant *Staphylococcus aureus* with a minimum inhibitory concentration of 62.5 µg/mL, and its disinfectant capacity was comparable to the potency of a commercial disinfectant. Moreover, 7α-acetoxy-6β-hydroxyroyleanone inhibits 5-LO with IC_50_ values of 1.3 and 5.1 µg/mL in cell-free and cell-based assays, respectively. These findings rationalize the ethnopharmacological use of *P. zeylanicus* as antimicrobial and anti-inflammatory remedy.

## 1. Introduction

*Plectranthus zeylanicus* Benth [synonym: *Coleus zeylanicus* (Benth)], locally known as Iruveriya [1], is a perennial herb within the family Lamiaceae, which has been widely employed in Ayurvedic and folk medicine in Sri Lanka for hundreds of years. Our previous investigations revealed potent 5-lipoxygenase (5-LO) inhibitory activities in lipophilic extracts of this plant, thus supporting its traditional usage as an anti-inflammatory remedy [2]. Leukotrienes (LT) are bioactive lipid mediators produced from arachidonic acid by 5-LO during the onset of infections with pathogenic bacteria such as *Escherichia coli* and *Staphylococcus aureus* [3], and they mediate typical symptoms of inflammation such as swelling, redness and pain [4]. LT also contribute to the development of chronic inflammatory disorders [5]. 

Apart from its utility for the treatment of inflammatory conditions, *P. zeylanicus* is also known as a medication for microbial infections, such as dysentery and diarrhea and also against smallpox, the common cold and eye diseases [6]. However, only a limited number of scientific reports are available to reinforce these traditional claims. For example, the essential oils of the plant displayed antimicrobial activity against *Proteus vulgaris*, *Aspergillus parasiticus*, *Aspergillus niger*, *Rhizopus oryzae* and *Colletotrichum musae* [7], while very mild antibacterial and antifungal activities with relatively high minimal inhibitory concentration (MIC) values up to 250 mg/mL were observed in the chloroform and ethanol extracts [8]. 

Previous studies on the phytoconstituents of *P. zeylanicus* led to the isolation of diterpenoids such as 7β-acetoxy-6β-hydroxyroyleanone, 7β-,6β-dihydroxyroyleanone, and 7α-acetoxy-6β-hydroxyroyleanone from its methanolic extracts [9]. In addition, geraniol, geranyl acetate, caryophyllene, eudesm-7(11)-en-4-ol, p-cymene, fenchyl acetate, fenchyl formate, and bornyl acetate were identified in the essential oils obtained from the aerial and the roots of *P. zeylanicus* grown in Sri Lanka [10], whereas about 80 compounds were detected in the essential oils of this plant grown in India [11]. Nevertheless, in-depth investigations on the bioactivity of secondary metabolites isolated from *P. zeylanicus* of Sri Lankan origin are not available to rationalize its therapeutic significance as an anti-inflammatory and antimicrobial remedy. Thus, in continuation of our early attempts of correlating the anti-inflammatory activity with the phytochemicals present in the lipophilic extracts of *P. zeylanicus* that inhibit 5-LO [2], the present study specifically focused on the determination of the antimicrobial activity of different extracts of this plant and to conduct bioactivity-guided fractionation to obtain the active secondary metabolites with antimicrobial, disinfectant and anti-inflammatory activities. 

In the present study, the anti-inflammatory activity of the isolated phytoconstituents was evaluated by analyzing their inhibitory activity on 5-LO in vitro. The antibacterial activity was determined against Gram-positive and Gram-negative bacteria, including methicillin-resistant *Staphylococcus aureus* (MRSA), while the antifungal activity was evaluated against *Candida albicans*. In order to determine the suitability of the antimicrobial compounds as potent disinfectants, the disinfectant potency was compared with that of a commercial disinfectant employing a surface disinfectant assay. Together, our data provide valuable insights into the ethnopharmacological significance of *P. zeylanicus* as a remedy for inflammatory disorders and microbial infections.

## 2. Results

### 2.1. Antimicrobial Activity of the Crude Extracts of P. zeylanicus

Our screening approach revealed that different extracts of *P. zeylanicus* possess antimicrobial activities against *S. aureus*, *S. saprophyticus* (Figure 1), *E. faecalis*, *S. typhi* and *P. aeruginosa* to different degrees; however, none of these extracts were active against *E. coli* and *S. flexneri*. The dichloromethane (DCM) extract displayed potent and broad-range activity against the tested bacterial species in comparison to the other extracts (Table 1), thus an initial phytochemical screening was performed and, thereafter, it was subjected to bioactivity-guided fractionation.

### 2.2. Phytochemical Screening by Gas Chromatography-Coupled Mass Spectrometric (GC-MS) Analysis

The GC-MS analysis of the DCM extract of *P. zeylanicus* led to the identification of several components based on the comparison of the obtained experimental mass spectra with those recorded in the NIST MS Search 2.0, Adams mass spectrum library, and also by comparison with those of the respective standards. Some of these compounds (hexadecanoic acid, 9,12,15-octadecatrienoic acid, stigmasterol, sitosterol and amyrin) have also been detected in the bioactive fractions of the *n*-hexane extract of this plant (Figure 2) as reported in our earlier publication [2]. In addition, the presence of several other diterpenes along with an intense peak representing a compound with the chemical formulae C_20–23_H_x_O_2–4_ was detected in the DCM extract. The compounds that have been tentatively identified are listed in Table 2.

### 2.3. Bioactivity-Guided Fractionation

In order to obtain deeper insights into the identity of the compounds responsible for the potent antimicrobial activity, the DCM extract was fractionated by column chromatography using ethyl acetate, *n*-hexane, and methanol as solvents into eleven fractions. These fractions were analyzed for antimicrobial activity against *S. aureus* and *S. saprophyticus* at 1000 µg/mL by the disc diffusion method. Out of the eleven fractions (F) of the DCM extract, F-5 (15% EtOAc in *n*-hexane), F-6 (25% EtOAc in *n*-hexane), F-7 (35% EtOAc in *n*-hexane) and F-8 (50% EtOAc in *n*-hexane) displayed antimicrobial activity against both bacterial species, with the highest activity observed in F-5 and F-6 (Table 3). Since analysis by thin-layer chromatography (TLC) indicated identical spots in these two fractions, they were pooled together and further separated by a Sephadex column packed with methanol. This led to the separation of five major sub-fractions (SF), and the most bioactive sub-fraction was further purified by silica gel column chromatography using *n*-hexane and EtOAc as the solvent system. This resulted in a pure compound, and the NMR and MS data analysis of this isolated compound revealed its identity as 7α-acetoxy-6β-hydroxyroyleanone (AHR) (Figure 3), which is an abietane type diterpenoid with the chemical formula of C_22_H_30_O_6_ (*m*/*z* = 391.213).

The ^1^H NMR and ^13^C NMR data of this compound were in agreement with those previously reported for 7α-acetoxy-6β-hydroxyroyleanone [12,13,14,15]. Independently from literature data, chemical shift assignment was achieved by 1D NMR (^1^H, ^13^C, APT) and 2D homo- and heterocorrelation experiments (^1^H-^1^H COSY, ^1^H-^1^H ROESY, HSQC and HMBC). The ^1^H and ^13^C NMR data of 7α-acetoxy-6β-hydroxyroyleanone are shown in Appendix A along with most recently reported NMR data of this compound for comparison [15].

### 2.4. Bioactivity Studies of the Isolated Compound

#### 2.4.1. Antimicrobial Activity and Disinfectant Potency

The isolated compound, 7α-acetoxy-6β-hydroxyroyleanone, displayed potent antibacterial activity against *S. saprophyticus* with a MIC of 31.5 µg/mL. Moreover, the MIC values of this compound against other Gram-positive bacteria, i.e., *S. aureus*, *E. fecalis* and nine clinical isolates of MRSA were found to be 62.5 µg/mL while only a moderate antibacterial activity was observed against the Gram-negative *S. typhi* and *P. aeruginosa* with MIC of 500 and 250 µg/mL, respectively. In addition, moderate antifungal activity was observed for this compound against *C. albicans,* with a MIC of 1000 µg/mL.

Furthermore, this compound exhibited a significant disinfectant activity against *S. aureus*, MRSA isolates and *P. aeruginosa*. The statistical analysis revealed that the compound was highly effective in reducing the mean colony counts of all the tested bacteria on rough and smooth surfaces when compared to the untreated surface (*p* < 0.01). The mean colony counts of all the organisms were more or less equal to that of the commercial disinfectant (used as the reference) on both types of surfaces (Table 4).

#### 2.4.2. Anti-Inflammatory Activity

Since a significant anti-inflammatory activity has been reported for the DCM extract of *P. zeylanicus*, that is, inhibition of 5-LO activity with IC_50_ of 1.2 and 12 µg/mL in cell-free and cell-based activity assays, respectively [2], 7α-acetoxy-6β-hydroxyroyleanone was evaluated for its activity against 5-LO. 7α-Acetoxy-6β-hydroxyroyleanone suppressed 5-LO activity in a cell-free assay using isolated human recombinant 5-LO with IC_50_ of 1.3 μg/mL and also inhibited 5-LO product formation in human neutrophils challenged with A23187 plus exogenous arachidonic acid with IC_50_ = 5.1 μg/mL (Figure 4). The reference 5-LO inhibitor zileuton blocked 5-LO activity with IC_50_ = 0.55 µM, as expected (not shown).

## 3. Discussion

In this study, attempts were made to identify bioactive secondary metabolites from the medicinal plant *P. zeylanicus* to rationalize and validate its traditional use as an anti-inflammatory and antimicrobial remedy. Our previous investigations revealed potent 5-LO inhibitory activity in the *n*-hexane and DCM extracts of this plant, thus supporting the traditional claims on its anti-inflammatory potential [2]. In the present study, a potent antibacterial activity was observed in the DCM extract of *P. zeylanicus*, and the bioactivity-guided fractionation revealed 7α-acetoxy-6β-hydroxyroyleanone as an active ingredient that moreover, efficiently inhibited 5-LO. Therefore, our results disclose potent antibacterial, disinfectant and anti-inflammatory activity for 7α-acetoxy-6β-hydroxyroyleanone and, thus, this rationalizes the ethnopharmacological use of *P. zeylanicus*.

With the increasing prevalence of multi-drug resistant microbial strains, which is considered as one of the leading causes for mortality and morbidity in hospitalized patients [16], the development of novel antimicrobial agents with diverse structures and mechanisms of action are highly demanding [17]. Since the therapeutic use of medicinal plants for the treatment of infectious diseases is as old as human civilization and has evolved along with it, the secondary metabolites present in the medicinal plants warrant exploration as promising candidates of novel antimicrobial agents [18]. In fact, a vast number of medicinal plants and the secondary metabolites thereof have been investigated for their antimicrobial potency [18,19,20]. The present investigation revealed that 7α-acetoxy-6β-hydroxyroyleanone displays strong antibacterial activity against various clinical isolates of MRSA with MIC of 62.5 µg/mL. Although the antimicrobial activity of 7α-acetoxy-6β-hydroxyroyleanone and its synthetic derivatives have been previously documented [15,21,22], none of these studies accomplished a potential application of this property. Since poor infection control practices in the hospital settings facilitate the transmission of drug-resistant bacteria among crowded hospital populations, appropriate environmental hygienic practices are extremely useful for hospital infection control. Hence, routine cleaning of high-touch surfaces or objects with disinfectants is adopted as one of the important preventive measures against the transmission of hospital-acquired infections [23]. Thus, special emphasis was given in our study to evaluate the disinfectant potency of 7α-acetoxy-6β-hydroxyroyleanone. It is noteworthy that the disinfectant capacity of 7α-acetoxy-6β-hydroxyroyleanone was comparable to the commercial disinfectant used as the positive control. To the best of our knowledge, this is the first report that describes the disinfectant potency of a plant secondary metabolite being equally effective as the commercial disinfectant, which stimulates the prospective development of eco-friendly biodegradable disinfectants. The recently developed protocol by Fonseka et al. [24] for in vitro mass propagation of *P. zeylanicus* might be helpful for a sustainable supply of plant material to isolate this bioactive compound and to develop a commercial product.

In addition, the 5-LO inhibitory action of 7α-acetoxy-6β-hydroxyroyleanone was disclosed for the first time by this study. Although several compounds such as β-caryophyllene [25], α-caryophyllene [26], α-tocopherol [27] and amyrin [28], with already proven anti-inflammatory activities, were detected in the DCM extract of *P. zeylanicus*, a correlation between the 5-LO inhibitory potency of the DCM extract and the presence of 7α-acetoxy-6β-hydroxyroyleanone can be proposed. The IC_50_ values in the cell-free and cell-based assays obtained for the DCM extract (IC_50_ = 1.2 and 12 µg/mL, respectively [2] were comparable to those of the 7α-acetoxy-6β-hydroxyroyleanone (IC_50_ = 1.3 and 5.1 µg/mL). Moreover, the potency of 7α-acetoxy-6β-hydroxyroyleanone against 5-LO in cell-free assays (IC_50_ of 1.3 µg/mL that corresponds to 3.3 µM) is even slightly superior over that of other well-known 5-LO inhibitory triterpenes like 3-O-acetyl-11-keto-β-boswellic acid (IC_50_ = 8 μM) and abietic acid (IC_50_ = 29.5 μM) or sesquiterpenes such as chamazulene (IC_50_ = 10 μM) [29]. Thus, 7α-acetoxy-6β-hydroxyroyleanone can be added to the list of plant-derived 5-LO inhibitors with high potency against this pro-inflammatory enzyme.

## 4. Materials and Methods

### 4.1. Plant Material

*P. zeylanicus* plants were collected in Nittambuwa (Western Province of Sri Lanka) in 2015/2016 and were identified by one of the authors (MN), who is a botanist. The morphological descriptions given in the books “A Revised Handbook to the Flora of Ceylon: volume-III, M.D. Dassanayake & F.R. Fosberg” and “Medicinal plants (indigenous and exotic) used in Ceylon: Volume 2 by D.M.A. Jayaweera” were used to confirm the identity of the plant. The plant was authenticated by comparison with the herbarium specimens at the National herbarium, Royal Botanical Garden, Peradeniya, Sri Lanka. A voucher specimen (Plec-WP-1603) was deposited at the Department of Biochemistry, Faculty of Medicine, University of Ruhuna, Sri Lanka for future reference.

### 4.2. Preparation of Crude Extracts

The plant material (whole plant) was thoroughly washed and dried in shade (30 ± 2 °C) for one week. Dried plants were powdered using a domestic grinder (Singer model KA-MIXEE). Three hundred and eighty grams of powdered material was successively extracted with 1800 mL of *n*-hexane, dichloromethane (DCM), ethyl acetate (EtOAc), and methanol (Roth, Karlsruhe, Germany) at room temperature using a linear shaker for 40 min. The extracts were evaporated to dryness with the use of a rotary evaporator (BÜCHI, R-114, Essen, Germany).

### 4.3. Evaluation of Antimicrobial Activity for Bioactivity-Guided Fractionation

The crude extracts from above were subjected to a preliminary screening for antimicrobial activity against standard isolates of *Staphylococcus aureus* (ATCC 25923) and *Escherichia coli* (ATCC 25922), as well as clinical isolates of *Staphylococcus saprophyticus*, *Enterococcus faecalis*, *Salmonella typhi*, *Pseudomonas aeruginosa* and *Shigella flexneri* available at the Department of Microbiology, Faculty of Medicine, University of Ruhuna, Sri Lanka. The disc diffusion method was employed for the pre-screening of extracts for antimicrobial activity and the MIC of the active extracts was determined by the broth-microdilution method.

#### 4.3.1. Pre-Screening of Crude Extracts for Antibacterial Activity

The crude extracts were initially screened for the antimicrobial activity at a concentration of 1000 µg/mL by the disc diffusion method. A loop full from an isolated colony of one-day-old cultures of each organism grown on blood agar was dissolved in normal saline and the turbidity of the solution was adjusted to the McFarland 0.5 standard. Mueller Hinton Agar (MHA) plates were inoculated with a suspension of each organism and thereafter the plates were allowed to dry. Then, filter paper discs (6 mm in diameter), containing the test extract/reference drugs were placed on the agar surface. The plates were incubated at 37 °C overnight and the zone of inhibition around each paper disc was measured. The zones ≥6 mm were considered as inhibition, resulting from significant antibacterial activity. The assay was conducted in triplicate. Gentamicin was used as the reference (positive control), while DCM was used as the negative control. The experiment was performed in duplicate.

#### 4.3.2. Determination of Minimum Inhibitory Concentration

Based on the preliminary observations, the active extracts were further subjected to the broth microdilution assay following the method described by Napagoda et al. [29] for the determination of the MIC values.

Briefly, 100 µL of solvent controls and test samples were added to the first wells of the microplate starting with a concentration of 2 mg/mL and then two-fold serially diluted down the wells. A part of the diluted culture (100 µL) with a turbidity standard equal to 0.5 McFarland and 50 µL of Muller Hinton Broth (MHB) were added to all wells. The microtiter plates were incubated for 24 h at 37 °C. After incubation, the absorbance was measured by a microplate reader (Biotek, ELx800, Minooski, VT, USA). The MIC was determined as the lowest concentration of test agent that prevents the visible growth of a bacterium. MBC (minimum bactericidal concentration) is the lowest concentration of an antibacterial agent required to kill a particular bacterium and was determined by sub-culturing the content of the above microtiter plate wells in agar plates. Gentamicin was used as the reference drug (positive control) for the assay. The assay was conducted in triplicate.

### 4.4. Phytochemical Profiling and Bioactivity-Guided Fractionation of Most Active Extract

Based on the results of the antimicrobial screening assays, the DCM extract was selected for the initial phytochemical screening and was further subjected to the bioactivity-guided fractionation.

#### 4.4.1. Phytochemical Profiling by GC-MS Analysis

GC-MS analysis was carried out by the method described by Napagoda et al. [30]. Dried crude DCM extract was dissolved in ethyl acetate (1 mg/mL) and analyzed on a gas chromatograph HP6890 (Agilent, Santa Clara, CA, USA) connected to an MS02 mass spectrometer from Micromass (Waters, Manchester, UK) with EI 70 eV using ZB5ms column (30 m × 0.25 mm, 0.25-μm film thickness; Phenomenex). The carrier gas was helium at a flow rate of 1 mL/min. The injector temperature was kept at 250 °C and the temperature program was set as 100 °C (2 min), 15 °C/min to 200 °C, 5 °C/min to 305 °C (20 min).

#### 4.4.2. Bioactivity-Guided Fractionation

DCM extract (430 mg) was subjected to silica gel column chromatography (Roth Kieselgel 60, 0.04–0.063 mm, 230–400 mesh). The sample was eluted with *n*-hexane, different mixtures of EtOAc in *n*-hexane (3%, 5%, 10%, 15%, 25%, 35%, 50%, 75%, 100%) and methanol sequentially to obtain eleven fractions.

The antimicrobial activity of the resulting fractions was determined against *S. aureus* and *S. saprophyticus* by the disc diffusion method. Based on the diameter of the zone of inhibition, the most active fraction was selected, and further fractionation was carried out by Sephadex LH-20 (Sigma Aldrich, Steinheim am Albuch, Germany) and silica gel column chromatography to isolate a pure compound. The antimicrobial, disinfectant and anti-inflammatory activities of this compound was determined.

The structure elucidation was achieved by ^1^H and ^13^C NMR spectroscopy and mass spectrometry. NMR spectra (^1^H NMR, ^13^C NMR, ^13^C APT, ^1^H-^1^H COSY, ^1^H-^1^H ROESY, ^1^H-^13^C HSQC and ^1^H-^13^C HMBC) were recorded on a Bruker Avance III HD 500 NMR spectrometer (Bruker Biospin, Karlsruhe, Germany) operating at 500.13 MHz for ^1^H and 125.75 MHz for ^13^C. The spectrometer was equipped with a 5 mm TCI cryoprobe. Standard Bruker pulse programs as implemented in Bruker TopSpin 3.5 were used for acquisition. The residual solvent signals of acetone-*d_6_* (^1^H-δ 2.04, ^13^C-δ 29.8) were used for referencing spectra in the ^1^H and ^13^C dimensions.

Mass spectrometric data were obtained by QExactive-HF-X and an Ultimate 3000 series RSLC (Dionex, Sunnyvale, CA, USA) chromatography system using electrospray ionization.

### 4.5. Evaluation of the Bioactivity of Pure Compound

#### 4.5.1. Antimicrobial Activity

##### Antibacterial Activity

The antibacterial activity and the MIC of the purified compound was determined by the broth microdilution method against *S. aureus*, *S. saprophyticus*, *E. faecalis*, *S. typhi*, *P. aeruginosa* and nine clinical isolates of methicillin-resistant *S. aureus* (MRSA).

##### Anti-Candida Activity

The in vitro anti-*Candida* activity of the isolated compound was assessed by the agar well diffusion method in Sabouraud Dextrose Agar (SDA). *Candida albicans* (ATCC 10231) was obtained from the Department of Microbiology, Faculty of Medicine, University of Ruhuna, Sri Lanka, and a fungal suspension matching with the McFarland 0.5 turbidity was prepared using isolated colonies of 48 h old cultures. Wells (diameter-6 mm, depth-5 mm) were prepared on SDA plates using a sterilized cork borer and the suspensions of the organisms were inoculated on these plates using a sterile cotton swab. The wells were filled with 50 µL of each of the test solutions (at 1000, 500, 250 and 125 µg/mL of the isolated compound, positive and negative controls) separately. Thereafter, the plates were incubated at 37 °C for 24 h and the zone of inhibition around each well was measured. The zones ≥6 mm were considered as inhibition resulting from significant antifungal activity. Fluconazole was used as the reference antifungal drug and DMSO was used as the negative control. The experiments were performed in triplicates and the diameter of the zone of inhibition was expressed as mean ± SD.

##### Disinfectant Potency

The disinfectant potency was evaluated against *S. aureus*, *P. aeruginosa* and three clinical isolates of MRSA following the method described by Napagoda et al. [31]. Pre-autoclaved rough (floor tile) and smooth (stainless steel) surfaces (50 cm^2^) were treated evenly with 1 mL of the bacterial suspension (equal to McFarland 0.5 turbidity) and allowed to dry for 1 h. On each surface, two squares (25 cm^2^) were labeled and the test solution (2.5 mL at its MBC concentration dissolved in 1% DMSO in water) was applied by a sterile cotton gauge in one square, while the other (labeled as non-disinfected area) was left without any treatment. After a contact period of 10 min, both areas were swabbed and each swab was vortexed in a tube containing 5 mL of MHB and a dilution was prepared as 1:10. Five drops of the dilution were inoculated on MHA plates and incubated for 48 h. A commercial disinfectant (Lifebuoy^®^ soap solution; Unilever, London, UK containing silver as the active ingredient) was used as the positive control. The experiment was performed in duplicates. Mean colony counts of different microorganisms on different surfaces were obtained, and the data were analyzed by one-way ANOVA with post hoc multiple comparisons. A *p*-value < 0.05 (*) was considered significant.

#### 4.5.2. Anti-Inflammatory Activity

##### Cell-Free 5-LO Activity Assay

*E. coli* (BL21) was transformed with pT3-5-LO plasmid, and recombinant 5-LO protein was expressed at 30 °C as described [31]. Cells were lysed in 50 mM triethanolamine/HCl pH 8.0, 5 mM EDTA, 1 mM phenylmethanesulphonyl fluoride, soybean trypsin inhibitor (60 µg/mL), and lysozyme (1 mg/mL), homogenized by sonication (3 × 15 s), and centrifuged at 40,000× *g* for 20 min at 4 °C. The 40,000× *g* supernatant (S40) was applied to an ATP-agarose column to partially purify 5-LO as described [32]. Aliquots of semi-purified 5-LO (0.5 µg) were diluted with 1 mL ice-cold PBS containing 1 mM EDTA. Samples were pre-incubated with the test compound or vehicle (0.1% DMSO). After 10 min at 4 °C, samples were pre-warmed for 30 s at 37 °C, and 2 mM CaCl_2_ plus 20 µM arachidonic acid were added to start the formation of 5-LO products. After 10 min, the reaction was stopped by the addition of one volume of ice-cold methanol, and the formed 5-LO products were analyzed by RP-HPLC as described [33]. 5-LO products include the all-trans isomers of LTB_4_ (tr-LTB_4_ isomers) as well as 5(*S*)-hydroperoxy-6-*trans*-8,11,14-*cis*-eicosatetraenoic acid (5-HPETE) and its corresponding alcohol 5(*S*)-hydroxy-6-*trans*-8,11,14-*cis*-eicosatetraenoic acid (5-HETE).

##### Cell-Based 5-LO Activity Assay

Neutrophils were isolated from peripheral blood (University Hospital Jena, Germany) which was taken from healthy adult volunteers with consent by venipuncture in heparinized tubes (16 IE heparin/mL blood). The blood donors had not taken any anti-inflammatory drugs within the prior 10 days. The blood was centrifuged at 4000× *g* for 20 min at 20 °C for preparation of leukocyte concentrates, which were then subjected to dextran sedimentation and centrifugation on lymphocyte separation medium (LSM 1077, PAA, Colbe, Germany). Contaminating erythrocytes of pelleted neutrophils were removed by hypotonic lysis. Neutrophils were then washed twice in ice-cold PBS pH 7.4 (PBS) and finally resuspended in PBS containing 1 mg/mL glucose or in PBS containing 1 mg/mL glucose plus 1 mM CaCl_2_ (PGC buffer) (purity > 96–97%). Neutrophils (5 × 10^6^) were resuspended in 1 mL PGC buffer, preincubated for 15 min at 37 °C with test compounds or vehicle (0.1% DMSO), and incubated for 10 min at 37 °C with 2.5 µM Ca^2+^-ionophore A23187 plus 20 µM arachidonic acid. The reaction was stopped on ice by the addition of 1 mL of methanol, and 30 µL 1 N HCl, 500 µL PBS, and 200 ng prostaglandin B_1_ were added. The samples were subjected to solid-phase extraction on C18-columns (100 mg, UCT, Bristol, PA, USA) and 5-LO products (LTB_4_, tr-LTB_4_ isomers, and 5-HETE) were analyzed by RP-HPLC. Quantities were calculated on the basis of the internal standard PGB_1_. Cysteinyl-LTs C_4_, D_4_ and E_4_ were not detected (amounts were below detection limit), and oxidation products of LTB_4_ were not determined.

##### Statistical Analysis

Data were expressed as mean ± S.E.M. The IC_50_ values were calculated from averaged measurements at four to five different concentrations of the compounds by nonlinear regression using GraphPad Prism software (San Diego, CA, USA) one site binding competition. Statistical evaluation of the data was performed by one-way ANOVA followed by a Bonferroni or Tukey-Kramer post-hoc test for multiple comparisons, respectively. A *p*-value < 0.05 (*) was considered significant.

## 5. Conclusions

Taken together, 7α-acetoxy-6β-hydroxyroyleanone was isolated from a DCM extract of *P. zeylanicus* and identified as a potent antibacterial and disinfectant agent that also displays efficient 5-LO inhibitory activity. These results further rationalize the traditional use of *P. zeylanicus* as natural pharmacon in Sri Lanka for the treatment of inflammatory conditions and microbial infections. Particularly, the extremely high disinfectant capacity exhibited by this compound could be considered as a signpost of an effective herbal disinfectant for years to come.

## Figures and Tables

**Figure 1 pharmaceuticals-15-00436-f001:**
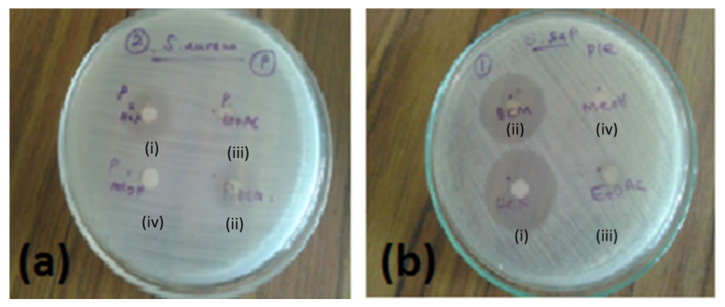
Antibacterial activity of different extracts of *P. zeylanicus* at 1000 µg/mL against (**a**) *S. aureus* or (**b**) *S. saprophyticus* where (i) *n*-hexane, (ii) DCM, (iii) EtOAc, or (iv) methanol were used for extraction. Photos are representatives out of *n* = 2 experiments with similar results.

**Figure 2 pharmaceuticals-15-00436-f002:**
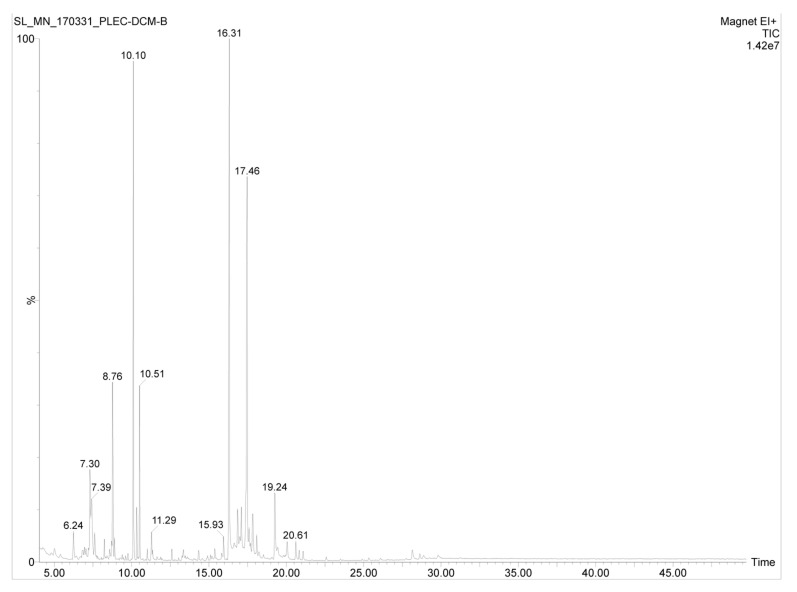
Total ion chromatogram of the DCM extract of *P. zeylanicus*.

**Figure 3 pharmaceuticals-15-00436-f003:**
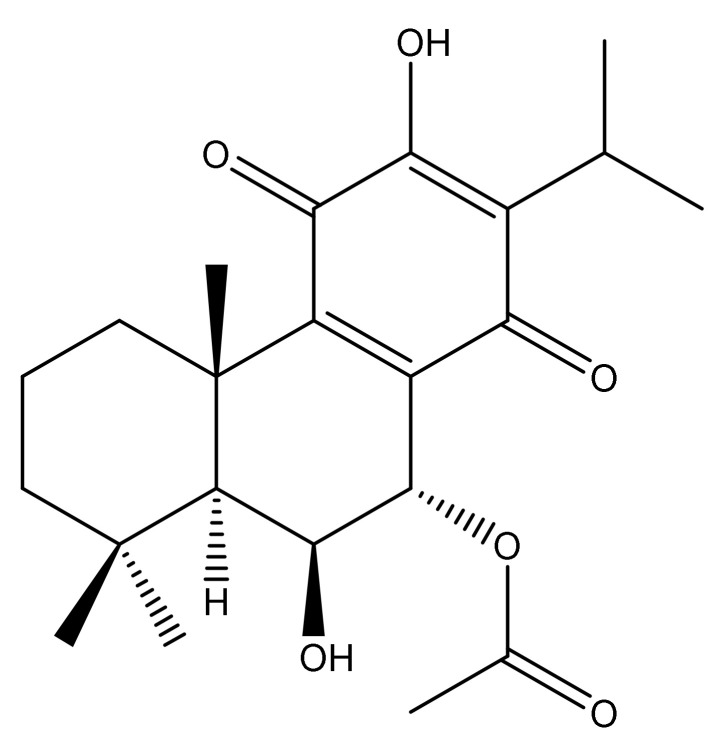
Chemical structure of 7α-acetoxy-6β-hydroxyroyleanone isolated after bioactivity-guided fractionation from the DCM extract of *P. zeylanicus*.

**Figure 4 pharmaceuticals-15-00436-f004:**
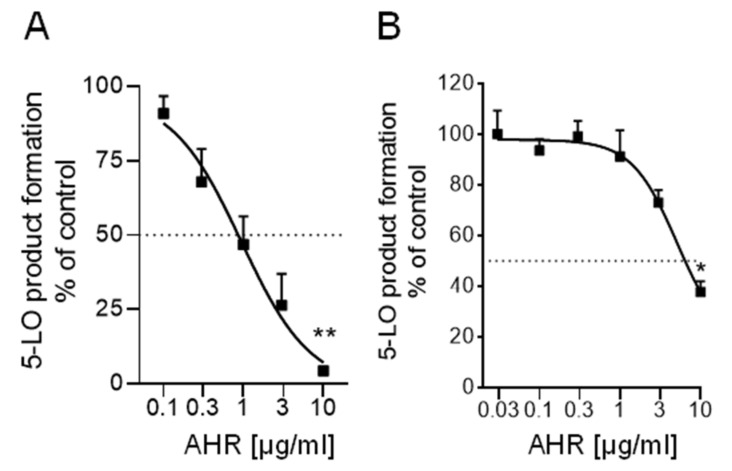
Inhibition of 5-LO by 7α-acetoxy-6β-hydroxyroyleanone (AHR) in (**A**) a cell-free assay and (**B**) in intact human neutrophils. Data are given as mean ± S.E.M, n = 4. Statistical evaluation of the data was performed by one-way ANOVA, * *p* < 0.05, ** *p* < 0.01.

**Table 1 pharmaceuticals-15-00436-t001:** MIC values of different extracts prepared from *P. zeylanicus* against various bacterial species. Data are given as means, *n* = 3.

Extract	MIC (μg/mL)						
	*S. aureus*	*S. saprophyticus*	*E. feacalis*	*S. typhi*	*P. aeruginosa*	*E. coli*	*S. flexneri*
*n*-hexane	31.25	31.25	62.5	500	-	-	-
DCM	31.25	31.25	125	500	250	-	-
ethyl acetate	250	500	-	1000	-	-	-
methanol	500	500	250	1000	-	-	-
Gentamicin (reference drug)	7.5	7.5	31.25	3.72	3.72	7.5	31.25

**Table 2 pharmaceuticals-15-00436-t002:** Tentatively identified compounds in the DCM extract of *P. zeylanicus*.

Retention Time (min)	Compound
6.24	a monoterpene acetate
6.96	β-caryophyllene
7.05	α-caryophyllene
7.30	γ-selinene
7.39	valencene
7.61	7-epi-α-selinene
8.23	viridiflorol
8.76	globulol
8.89	α-bisabolol
10.10	neophytadiene
10.51	(3,7,11,15)-tetramethyl-2-hexadecen-1-ol
11.29	hexadecanoic acid
13.28	9,12-octadecadienoic acid
13.35	9,12,15-octadecatrienoic acid
13.60	octadecanoic acid
14.31	phytolacetate
16.31	stigmasta-5,24(28)-dien-3-ol
17.46	a compound with the formulae of C_20–23_H_x_O_2–4_
20.83	heptacosane
22.58	squalene
26.57	α-tocopherol
28.16	stigmasterol
28.87	sitosterol
29.81	amyrin

**Table 3 pharmaceuticals-15-00436-t003:** Antimicrobial activity of the different fractions (F) obtained by silica gel column chromatography of the DCM extract of *P. zeylanicus* and sub-fractions (SF) obtained from the combined fraction F-5 + F-6. The diameter of the zone of bacterial growth inhibition is given as means ± S.D, n = 2.

Fraction (F)/ Sub-Fraction (SF) (No.)	Diameter of the Zone of Growth Inhibition (mm) of	
	*S. aureus*	*S. saprophyticus*
F-1	-	6 ± 0
F-2	-	10 ± 0
F-3	-	-
F-4	-	-
F-5	18 ± 0	29 ± 0
F-6	19 ± 0	28 ± 0
F-7	10 ± 0	15 ± 0
F-8	7 ± 0	8 ± 0
F-9	-	6 ± 0
F-10	-	-
F-11	-	-
SF-1	-	-
SF-2	10 ± 0	19 ± 0
SF-3	22 ± 0	33 ± 0
SF-4	-	-
SF-5	-	-
gentamicin (reference drug)	16 ± 0	22 ± 0

**Table 4 pharmaceuticals-15-00436-t004:** Comparison of mean colony counts of different microorganisms on different surfaces by one-way ANOVA with post hoc multiple comparisons.

	Comparison of Mean Colony Counts of Different Microorganisms on Different Surfaces											
	*P. aeruginosa*				*S. aureus*				MRSA			
	Smooth Surface		Rough Surface		Smooth Surface		Rough Surface		Smooth Surface		Rough Surface	
ANOVA Test Result	*p* = 0.021		*p* = 0.001		*p* = 0.001		*p* = 0.001		*p* = 0.000		*p* = 0.000	
	Mean Colony Count	Post Hoc Test vs. Untreated	Mean Colony Count	Post Hoc Test vs. Untreated	Mean Colony Count	Post Hoc Test vs. Untreated	Mean Colony Count	Post Hoc Test vs. Untreated	Mean Colony Count	Post Hoc Test vs. Untreated	Mean Colony Count	Post Hoc Test vs. Untreated
Untreated surface	5.75		7.5		45.25		10.75		487.5		450.0	
Commercial disinfectant	1.5	*p* = 0.017	1.5	*p* = 0.001	0.5	*p* = 0.001	1.0	*p* = 0.001	5.0	*p* = 0.000	1.25	*p* = 0.000
Isolated compound	1.5	*p* = 0.017	1.0	*p* = 0.001	1.0	*p* = 0.001	1.0	*p* = 0.001	0.25	*p* = 0.000	6.25	*p* = 0.000

## Data Availability

The data presented in this study are available on reasonable request from the corresponding authors. The data are not publicly available due to privacy.

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
