# Peer review of "Plectranthus zeylanicus*: A Rich Source of Secondary Metabolites with Antimicrobial, Disinfectant and Anti-Inflammatory Activities"

_pharmaceuticals, 2022, doi:10.3390/ph15040436_

Round 1
Reviewer 1 Report
Minor remarks
Please, avoid the use of the first-person plural in the scientific manuscript. Use only the third-person singular.
Latin names of microorganisms should be presented in italics. Also, Greek symbols should be depicted in italics.
All other minor remarks are given in the manuscript.
Major remarks
Please, provide a review of the literature on recent references. There are many very old references.

Author Response
Minor remarks
Please, avoid the use of the first-person plural in the scientific manuscript. Use only the third-person singular.
AUTHOR: Done, we corrected this as requested.
Latin names of microorganisms should be presented in italics. Also, Greek symbols should be depicted in italics.
AUTHOR: Done, we corrected this as requested.
All other minor remarks are given in the manuscript.
AUTHOR: Done, we have carefully checked these corrections in the provided PDF of the reviewer and transferred these to the revised manuscript.
Major remarks
Please, provide a review of the literature on recent references. There are many very old references.
AUTHOR: A new more recent reference [24] was added. However, since there are only a few studies conducted on P. zeylanicus apart from the study done by Napagoda et al (2014), it is not possible to include recent references on the biological or pharmacological activities of this plant.
Reviewer 2 Report
This paper describes the separation of the compound from Plectranthus zeylanicus through a bio-activity guided fraction, and antimicrobial, disinfectant potency, and anti-inflammatory experiments performed.
*Some minor problems with this paper should be corrected before publication.
Suggestions and comments
Line 44: Iruveriya, [1] → Iruveriya [1], (position of comma)
Lines 88-91: Change the scientific names to the italic font.
Line 134: m/ z → m/z
Line 144: ~1D NMR → 1D NMR
Line 161: p<0.01 → p<0.01
Lines 354 and 397: p value → p value
Lines 355, 356, and 372: The numbering is wrong, authors need to change them.
Funding: revise “The APC was funded by XXX”
Appendix A: In the table, there are "~" and "*" marks in the Bernardes reference, please write what they mean.
Reference section.
Check the “instructions for Authors” –There is no need to write an “issue number” in the Reference section.
*As shown in Table 1 and Figure 1, the DCM fraction and n-hexane fraction show good results. Why did the authors choose DCM only? Even in Fig. 1(b), the result of n-hexane looks more effective.
*Has author checked the n-hexane or DCM fraction to see if there are other substances similar to the isolated 7α-acetoxy-6β-hydroxyoryleanone? Also, can the isolated compound be checked on GC-MS?
*Is there any possibility that the acetylation of the isolated compound is an artifact generated during the experiment?
Author Response
Comments and Suggestions for Authors
This paper describes the separation of the compound from Plectranthus zeylanicus through a bio-activity guided fraction, and antimicrobial, disinfectant potency, and anti-inflammatory experiments performed.
Some minor problems with this paper should be corrected before publication.
Suggestions and comments
Line 44: Iruveriya, [1] → Iruveriya [1], (position of comma)
AUTHOR: Corrected as requested.
Lines 88-91: Change the scientific names to the italic font.
AUTHOR: Corrected as requested.
Line 134: m/ z → m/z
AUTHOR: Corrected as requested.
Line 144: ~1D NMR → 1D NMR
AUTHOR: Corrected as requested.
Line 161: p<0.01 → p<0.01
AUTHOR: Corrected as requested.
Lines 354 and 397: p value → p value
AUTHOR: Corrected as requested.
Lines 355, 356, and 372: The numbering is wrong, authors need to change them.
AUTHOR: Corrected as requested.
Funding: revise “The APC was funded by XXX”
AUTHOR: Done, we have added the funder.
Appendix A: In the table, there are "~" and "*" marks in the Bernardes reference, please write what they mean.
AUTHOR: Corrected as requested, we now explain this, see below the table.
Reference section.
Check the “instructions for Authors” –There is no need to write an “issue number” in the Reference section.
AUTHOR: Corrected as requested.
As shown in Table 1 and Figure 1, the DCM fraction and n-hexane fraction show good results. Why did the authors choose DCM only? Even in Fig. 1(b), the result of n-hexane looks more effective.
AUTHOR: The DCM extract displayed a broader range of antimicrobial activity than the n-hexane extract, i.e. DCM extract inhibited the growth of S. aureus, S. saprophyticus, E. faecalis, S. typhi and P. aeruginosa, whereas the n-hexane extract inhibited only S. aureus, S. saprophyticus, E. faecalis and S. typhi and was inactive against P. aeruginosa. This is mentioned in the manuscript under 2.1 and thus, the DCM extract was selected for further studies. Moreover, although the zone of inhibition observed with the n-hexane extract against S. saprophyticus was bit larger than that observed with the DCM extract at a fixed concentration 1000 µg/mL (Fig. 1(b), the MIC was the same for both extracts against S. saprophyticus.
Has author checked the n-hexane or DCM fraction to see if there are other substances similar to the isolated 7α-acetoxy-6β-hydroxyoryleanone? Also, can the isolated compound be checked on GC-MS?
AUTHOR: Our previous findings revealed the presence of a similar compound, namely coleone P in the n-hexane extract (Napagoda et al. 2014). Although it may be possible to detect the presence of the isolated compound by GC-MS, we have not detected it under our experimental settings.
Is there any possibility that the acetylation of the isolated compound is an artifact generated during the experiment?
AUTHOR: This compound has been isolated from several Plectranthus species by different research groups under varying conditions, therefore, we are confident that it cannot be an artifact generated during the experiment.